# Transcriptome and Metabolome Profiling to Explore the Causes of Purple Leaves Formation in Non-Heading Chinese Cabbage (*Brassica rapa* L. ssp. *chinensis* Makino var. *mutliceps* Hort.)

**DOI:** 10.3390/foods11121787

**Published:** 2022-06-17

**Authors:** Ying Zhao, Xinghua Qi, Zeji Liu, Wenfeng Zheng, Jian Guan, Zhiyong Liu, Jie Ren, Hui Feng, Yun Zhang

**Affiliations:** Horticulture College, Shenyang Agricultural University, 120 Dongling Road, Shenhe District, Shenyang 110866, China; zhaoying941013@163.com (Y.Z.); qixinghua4858@163.com (X.Q.); liuzeji1315@163.com (Z.L.); zwf19980311@163.com (W.Z.); philg1212@163.com (J.G.); liuzhiyong99@syau.edu.cn (Z.L.); 2019500023@syau.edu.cn (J.R.); fenghuiaaa@syau.edu.cn (H.F.)

**Keywords:** anthocyanin, non-heading Chinese cabbage, metabolome, transcriptome

## Abstract

Purple non-heading Chinese cabbage is one of the most popular vegetables, and is rich in various health-beneficial anthocyanins. Research related to genes associated with anthocyanin biosynthesis in non-heading Chinese cabbage is important. This study performed integrative transcriptome and metabolome analysis in the purple non-heading Chinese cabbage wild type (WT) and its green mutant to elucidate the formation of purple leaves. The anthocyanin level was higher in purple than in green plants, while the contents of chlorophyll and carotenoid were higher in the green mutant than in the purple WT. Twenty-five anthocyanins were identified in purple and green cultivars; eleven anthocyanin metabolites were identified specifically in the purple plants. RNA-seq analysis indicated that 27 anthocyanin biosynthetic genes and 83 transcription factors were significantly differentially expressed between the WT and its mutant, most of them with higher expression in the purple than green non-heading Chinese cabbage. Transcriptome and metabolome analyses showed that UGT75C1 catalyzing the formation of pelargonidin-3,5-*O*-diglucoside and cyanidin-3,5-*O*-diglucoside may play a critical role in purple leaf formation in non-heading Chinese cabbage. Therefore, these results provide crucial information for elucidating the formation of purple leaves in non-heading Chinese cabbage.

## 1. Introduction

Non-heading Chinese cabbage (*Brassica rapa* L. ssp. *chinensis* Makino var. *mutliceps* Hort.) is one of the most widely consumed *Brassica* crops in the cruciferous family in the world. Purple varieties are excellent for human health because they are rich in anthocyanins—the purple color has been used as a quality parameter in non-heading Chinese cabbage breeding programs, and purple non-heading Chinese cabbage helps in the development of natural coloring matter or functional foods. The various colors in plants are decided by a few pigments—chlorophyll, carotenoids, and anthocyanins. Anthocyanins play a crucial role in leaf coloration; they are water-soluble pigments that are extensively distributed in plants, which belong to the compounds of the flavonoid class. Anthocyanins confer pink, red, purple, and blue colors to various tissues and organs in fruits, flowers, vegetables, and seeds [1,2,3]. Furthermore, anthocyanins reduce damage by cold, drought, salt, ultraviolet, and low-phosphate stresses, and boost pollination and seed dispersal by attracting animals and insects [4,5,6]. Recently, anthocyanins have been considered as a beneficial food ingredient due to their physiological and biological functions, including antiaging, antioxidant activity, and cancer prevention [7,8].

The pathway of anthocyanin biosynthesis is a process of secondary metabolism and has been well characterized in numerous species, including *Arabidopsis* (*Arabidopsis thaliana*), ornamental kale (*Brassica oleracea* L. var. *acephala*), and kohlrabi [9,10,11]. Studies on anthocyanin biosynthesis in non-heading Chinese cabbage are important. The anthocyanin biosynthetic pathway includes the phenylpropanoid pathway, early biosynthesis stage, and later biosynthesis stage. In brief, anthocyanins are synthesized from phenylalanine through the phenylpropanoid and flavonoid biosynthesis pathways in the cytoplasm, and then are transported to the vacuole after modification by methylation, acetylation, and glycosylation [12]. First, the conversion of phenylalanine to cinnamic acid is catalyzed by the phenylalanine ammonia-lyase (PAL) enzyme. Cinnamic acid is converted to dihydrokaempferol through a series of enzyme activities, including chalcone synthase (CHS), cinnamate 4-hydroxylase (C4H), chalcone isomerase (CHI), 4-coumaroyl CoA ligase (4CL), flavanone 3′-hydroxylase (F3′H), flavanone 3-hydroxylase (F3H), and flavanone 3′-5′-hydroxylase. Next, unmodified and colorless anthocyanins are converted from dihydrokaempferol by dihydroflavonol 4-reductase (DFR). Then, anthocyanidin synthase (ANS) catalyzes the conversion of the colorless anthocyanins to colored anthocyanins. The anthocyanins are modified through methylation, acetylation, and glycosylation by UDP-rhamnose anthocyanidin-3-glucoside rhamnosyl transferase, flavonoid-5-glucosyltransferase, flavonoid-3-glucosyltransferase, and flavonoid-7-glucosyltransferase. The modified anthocyanidins are transported from the cytoplasm to the vacuole by the ATP-binding cassette, glutathione transferase, and proteins of the toxic-compound-extrusion family [13,14,15].

Anthocyanin biosynthesis is regulated by multiple transcription factors (TFs). For example, MYB, bHLH, and WD40 form one transcriptional activation complex MBW for regulating the expression of the genes of anthocyanin biosynthesis, in which MYB plays the main role [16]. Two MYB types (R2R3- and R3-MYB) compete to generate the MBW complex, increasing or inhibiting anthocyanin biosynthesis, respectively [15,17]. Furthermore, independent R2R3-MYB could be involved in regulating anthocyanin biosynthesis [18]. bHLH TFs are significant regulators of anthocyanin biosynthesis, which regulate the expression of *DFR*, *UFGT*, and *CHS* by forming complexes with WD40 and MYB [19]. The WD40 protein regulates anthocyanin biosynthesis by interacting with MYB and bHLH [20]. LBD37, LBD38, LBD39, MYBL2, and CPC negatively regulate anthocyanin biosynthesis in *Arabidopsis* [21,22,23,24,25]. Recently, another transcription factor, WRKY, was also discovered to affect anthocyanin biosynthesis in plants [6]. In purple heading Chinese cabbage, two markers (CL-12 and B214-87) linked to the purple gene *BrPur* were obtained by the BSA method [26]. Anthocyanin biosynthesis may be activated by *BrMYB2* and *BrTT8* in purple heading Chinese cabbage [7]. In addition, *BcMYB44* and *BrMYBL2.1-G* were found to negatively regulate anthocyanin biosynthesis by inhibiting the expression of *F3H*, *DFR*, and *CHS* in Chinese cabbage [27,28]. The expression of genes related to anthocyanin biosynthesis is different in various species; thus, it is important to identify the gene expression involved in anthocyanin biosynthesis.

In this study, a purple DH line and its green spontaneous mutant with the same genetic background, except for the leaf color, were used for identifying different metabolites and differentially expressed genes (DEGs) in anthocyanin biosynthesis by the integrated analyses of metabolome and transcriptome in non-heading Chinese cabbage. The genes participating in chloroplast biosynthesis were also found between the WT and mutant of non-heading Chinese cabbage. qRT-PCR was used for verifying the results of the transcriptome analysis. The results of this study can provide new insights for understanding the formation of purple leaves in non-heading Chinese cabbage.

## 2. Materials and Methods

### 2.1. Plant Materials

The purple DH line 18sy026 was obtained from the ‘Purple non-heading Chinese cabbage strain’ by isolated microspore culturing. A green-leaf-color mutant was spontaneously mutated from the self-pollination of this purple DH line. Therefore, their genetic backgrounds were the same, except for their purple and green leaf colors. Therefore, the mutant and its wide type were ideal materials for transcriptomic analysis. The purple line (WT) and green mutant were cultivated in a greenhouse at Shenyang Agricultural University, Liaoning, China. The leaves used for the measurement of anthocyanin, metabolome study, RNA sequencing (RNA-Seq), and qRT-PCR validation were collected at the same time. Three similar purple WTs and green mutants were respectively selected, and leaves were collected from the same part of each plant to form three biological replicates.

### 2.2. Total Anthocyanins Analysis

Freeze-dried leaves were used for extracting chlorophyll and carotenoids referring to the method described by Li et al. [29]. The samples of non-heading Chinese cabbage (0.2 g) were cultured in 5 mL of 96% ethanol (*v*/*v*) for 24 h at 24 ◦C, then vortexed for 30 s after culturing for 12 h in the dark. The sample absorbances were measured at 663, 646, and 470 nm. The formulas for calculating the pigment content are as follows:$$\mathrm{Chlorophyll}\;a\;\mathrm{content}\;\left(\mathrm{mg}\cdot g^{-1}\mathrm{FW}\right)=\frac{\left(12.21\times A_{663}-2.81\times A_{646}\right)\times V}{1000\times W}\;\mathrm{Chlorophyll}\;b\;\mathrm{content}\;\left(\mathrm{mg}\cdot g^{-1}\mathrm{FW}\right)=\frac{\left(20.13\times A_{646}-5.03\times A_{663}\right)\times V}{1000\times W}\;\mathrm{Carotenoid}\;\mathrm{content}\;\left(\mathrm{mg}\cdot g^{-1}\mathrm{FW}\right)=\frac{\left(4.4\times A_{470}-0.01\times \mathrm{Chl}\;a-0.45\times \mathrm{Chl}\;b\right)\times V}{1000\times W}$$

The anthocyanin content was identified using the spectrophotometric pH differential method in non-heading Chinese cabbage as described by Dong et al. [19]. First, frozen leaves were mashed into a powder. Next, solution A (300 mM HCl and 100 mM KCl, and pH 1.0) and solution B (120 mM HCl and 200 mM sodium acetate, and pH 4.5) were prepared. Then 2 mL each of buffer A and B were separately applied to the extracted 100 mg of powder. Sequentially, the mixture was centrifuged at 13,000 rpm for 15 min at 4 °C. The absorbance of the supernatants at 510 nm was measured. The anthocyanin content was calculated according to the following formula:$$\mathrm{Anthocyanin}\;\mathrm{content}\;\left(\mathrm{mg}\cdot g^{-1}\mathrm{FW}\right)=\frac{\left(A1-A2\right)\times Mr\times 1000}{\varepsilon }$$
where *A*1 is the absorbance of the supernatant fluid collected from buffer A at 510 nm, *A*2 is the absorbance of the supernatant fluid collected from buffer B at 510 nm, *Mr* is the molecular mass of cyanidin-3-glucoside chloride (484.8), and *ε* represents the molar absorption coefficient at 510 nm (24,825). Three repeated tests were performed in the determination of the pigment content.

### 2.3. Extraction and Separation of Anthocyanins

Metabolites of anthocyanins were identified through Metware Biotechnology Co., Ltd. (Wuhan, China) according to the AB Sciex QTRAP 6500 LC-MS/MS platform. The frozen samples were crushed using a mixer mill (MM 400, Retsch, Haan, Germany) with one zirconia bead at 30 Hz for 90 s. Furthermore, 50 mg of powder was extracted using 0.5 mL of methanol/water/hydrochloric acid (500:500:1, *v*/*v*/*v*). Then, the extract was eddied for 5 min, subjected to ultrasound for 5 min, and centrifuged for 3 min at 12,000× *g* at 4 °C, and the procedure was repeated three times. Before liquid chromatography with tandem mass spectrometry (LC-MS/MS) analysis, the supernatant fluid was collected and filtrated (0.22 μm, Anpel, Shanghai, China).

The extracts were analyzed using an UPLC-ESI-MS/MS system (UPLC, SCIEX, Framingham, MA, USA), ExionLC™ AD (SCIEX, USA), https://sciex.com.cn/ (November 1^st^, 2021; MS, QTRAP^®^ 6500+, https://sciex.com.cn/ (1 November 2021). The samples were injected into a C18 column (ACQUITY BEH C18 1.7 µm, 2.1 mm × 100 mm). The binary solvent system used was ultrapure water containing 0.1% formic acid (Sigma-Aldrich, Darmstadt, Germany) as mobile phase A and methanol (Merck, Darmstadt, Germany) containing 0.1% formic acid as mobile phase B. The elution gradient of A:B (*v*/*v*) was 95:5 at 0 min, 50:50 at 6 min, 5:95 at 12 min, hold for 2 min, 95:5 at 14 min, and hold for 2 min. The flow rate was maintained at 0.35 mL/min, the column temperature was kept at 40 °C, and the injection volume was 2 μL.

### 2.4. Metabolite Identification and Quantification

The effluent was connected to one ESI–triple quadrupole-linear ion trap–MS/MS system (API 6500 Q TRAP). LIT and triple quadrupole scans were obtained on the triple quadrupole–linear ion trap mass spectrometer (Q TRAP), API 6500 Q TRAP UPLC/MS/MS System, which contained the ESI Turbo Ion-Spray interface. The system was operated in the positive ion mode and controlled using the Analyst 1.6.3 software (AB Sciex, Framingham, MA, USA). The operation parameters of the ESI source were: turbo spray, ion source; ion spray voltage (IS) 5500 V (positive ion mode); source temperature 550 °C; and curtain gas (CUR) 35 psi. The DP and CE of individual MRM transitions were further optimized. Based on the metabolites eluted during that period, a specific group of MRM transitions was monitored for each period. Qualitative analysis of the mass spectrometry data was carried out. Quantitative analysis was completed by the MRM of the triple quadrupole linear ion trap mass spectrometer according to the MWDB (Metware Database) database constructed by the standard. In the MRM, the four-stage rod first screened the precursor ions of the target substance. The precursor ions were broken to form multiple fragment ions after the ionization induced by the impact chamber, and the fragment ions were filtered through the triple four-stage rod to select the required characteristic fragment ions. The integral peak areas of all tested samples were substituted into the linear equation of the standard curve for calculation. Content of metabolites (μg/g) = c × V/1000000/m, where c: the concentration value (ng/mL) obtained by substituting the integral peak area in the sample into the standard curve, V: volume of solution used for extraction (μL), and m: weighed sample mass (g). Significantly different metabolites in terms of content were screened with the threshold of fold change (FC) ≥ 2 or ≤0.5.

### 2.5. RNA-Seq Analysis

The total RNA was extracted from the leaves using an RNA Extraction Kit (Aidlab, HaiDian, Beijing, China). The quantity of RNA was verified by the Agilent Bioanalyzer 2100 system (Agilent Technologies, Palo Alto, CA, USA). Randomly selected triplicate leaf samples of the purple WT and green mutant were used for constructing six cDNA libraries (Purple 1, Purple 2, Purple 3, Green 1, Green 2, and Green 3). Paired-end sequencing was carried out using an Illumina HiSeq 4,000 (Novogene Bioinformatics Technology Co. Ltd., Beijing, China). The low-quality sequences and adapters were filtered from the raw reads. However, clean reads were mapped to the Brassica reference genome (http://brassicadb.cn/#/ (30 August 2021)). The Fragments Per Kilo bases per Million fragments (FPKM) method was used for calculating the expression levels of genes. The DEGs were identified based on the following thresholds: *p* value < 0.05 and |log2(foldchange)| ≥ 1. The clusterProfiler R package was used for Gene Ontology (GO) (http://www.geneontology.org/ (5 September 2021)) and Kyoto Encyclopedia of Genes and Genomes (KEGG) (http://www.genome.ad.jp/kegg/ (5 September 2021)) enrichment analysis of DEGs.

### 2.6. qRT-PCR Analysis

The total RNA was extracted from the leaves using an RNA Extraction Kit (Aidlab, Beijing, China). The PCR primers are listed in the Appendix A. The actin gene was used as the internal control. All reactions of qRT-PCR were performed in 20 mL total volume containing 10 mL of 2 × UltraSYBR Mixture (Low ROX; CWBIO), 0.4 µL of specific primers (each), 0.8 µL of template cDNA, and 8.4 µL of ddH_2_O. All experiments were performed three times. The expression levels of thirty-one DEGs (*BraA04g026260.3C*, *BraA09g044270.3C*, *BraA04g006280.3C*, *BraA07g021160.3C*, *BraA07g031570.3C*, *BraA03g039690.3C*, *BraA05g025870.3C*, *BraA06g004850.3C*, *BraA10g024990.3C*, *BraA03g005990.3C*, *BraA02g005190.3C*, *BraA10g028200.3C*, *BraA09g046060.3C*, *BraA07g022020.3C*, *BraA09g042420.3C*, *BraA03g045490.3C*, *BraA10g030360.3C*, *BraA06g027070.3C*, *BraA09g019440.3C*, *BraA01g013470.3C*, *BraA03g050560.3C*, *BraA08g009740.3C*, *BraA01g032130.3C*, *BraA09g003850.3C*, *BraA08g035120.3C*, *BraA02g006880.3C*, *BraA10g022740.3C*, *BraA09g028560.3C*, *BraA09g013280.3C*, *BraA07g035710.3C*, and *BraA03g060820.3C*) related to anthocyanin biosynthesis and regulation were determined simultaneously. The relative expression was calculated via the 2^−∆∆Ct^ method.

### 2.7. Statistical Analysis

International Business Machines Statistical Package for Social Sciences version 20.0 (SPSS Inc., Chicago, IL, USA) was used for statistical analysis of variance. Data were presented as the mean ± standard errors for triplicates. Significance for differences was set at *p* < 0.05.

## 3. Results

### 3.1. Identifications of Phenotype and Pigment Content in Purple Non-Heading Chinese Cabbage and Its Green Mutant

Significant differences were found between the purple non-heading Chinese cabbage and the green mutant in terms of leaf color (Figure 1A). Visual inspection of the non-heading Chinese cabbage cultivars showed that the purple plant had a deeper purple pigmentation than the green mutant. There was a remarkable difference in the anthocyanin content between the purple WT and the green mutant. Higher anthocyanin accumulation (19.40 mg/g of FW) was identified in the purple non-heading Chinese cabbage than in the green mutant (1.23 mg/g) (Figure 1B). The contents of chlorophyll and carotenoids in the green plants were prominently higher than those in the purple plants (Figure 1C,D). The diversity in pigment content seems to make the leaves of non-heading Chinese cabbage display different colors.

### 3.2. Identification of Anthocyanin Metabolites from the Leaves of Non-Heading Chinese Cabbage

Leaf extracts from the leaves of the purple plant and green mutant were analyzed by MRM to detect anthocyanins, and twenty-five metabolites were identified from both non-heading Chinese cabbages (Table 1 and Appendix A). These metabolites were divided into seven groups, including four delphinidins, nine cyanidins, five petunidins, two pelargonidins, three peonidins, one malvidin, and one flavonoid (Figure 2). When setting FC ≥ 2 or ≤0.5 as the thresholds of significant differences, four metabolites were significantly different between the purple non-heading Chinese cabbage and its green mutant, respectively. The cyanidin-3,5-*O*-diglucoside, petunidin-3-*O*-sambubioside, and peonidin-3-*O*-glucoside increased by 5.73-, 2.72-, and 3.22-fold, respectively, in the purple leaves compared with the green leaves. Notably, the petunidin-3-*O*-glucoside level was markedly higher in the green mutant than that in the purple WT, showing that a small amount of anthocyanin was detected in the green mutant (Table 1). Additionally, twelve specific metabolites were identified between the purple non-heading Chinese cabbage and its green mutant, of which eleven anthocyanins (cyanidin-3-*O*-sambubioside-5-*O*-glucoside, cyanidin-3-*O*-sophoroside, cyanidin-3-*O*-sambubioside, cyanidin-3,5,3′-*O*-triglucoside, cyanidin-3-*O*-(coumaryl)-glucoside, cyanidin-3-(6-caffeoyl)-glucoside, pelargonidin-3,5-*O*-diglucoside, pelargonidin-3,5,3′-*O*-triglucoside, peonidin-3,5-*O*-diglucoside, peonidin-3,5,3′-*O*-triglucoside, and malvidin-3-*O*-(6-*O*-malonyl-beta-D-glucoside)) were only detected in the purple non-heading Chinese cabbage; delphinidin-3-*O*-(6-*O*-malonyl-beta-D-glucoside) was only identified in the green sample (Table 1).

### 3.3. Transcriptome Analysis of Purple Non-Heading Chinese Cabbage and Its Green Mutant

The cDNA libraries were constructed from leaves of the purple plant and its green mutant with triplicates for each sample and RNA-Seq analysis was carried out based on the Illumina HiSeq 4000 (Illumina, Santiago, CA, USA) platform to study the mechanism underlying anthocyanin accumulation in non-heading Chinese cabbage with different leaf colors. Therefore, a total of 52,178,406–60,045,254 (Purple 1–Purple 3) and 49,409,626–58,277,114 (Green 1–Green 3) raw reads were obtained from purple and green non-heading Chinese cabbage, respectively. Furthermore, after filtering out the adaptors and unknown or low-quality reads, 51,156,540–58,856,784 and 48,245,624–57,101,472 clean reads remained in Purple 1–3 and Green 1–3, respectively (Appendix A). At the Q30 level, >93% of the clean reads had Phred-like quality scores, indicating accurate sequencing. More than 74% of the clean reads were mapped to the Brassica reference genome in purple (Purple 1–3) and green (Green 1–3) non-heading Chinese cabbage (Appendix A).

Comparative analysis of the gene expression profiles was conducted to determine the DEGs between purple non-heading Chinese cabbage and the green mutant. With restrictive conditions of |log2 (fold-change)| > 1 and *p* value < 0.05, 1590 genes were identified as DEGs, including 724 up-regulated and 866 down-regulated genes in the green mutant compared with the WT. The number of up-regulated DEGs in green non-heading Chinese cabbage was significantly lower than that of down-regulated DEGs (Figure 3 and Appendix A).

### 3.4. GO and KEGG Analysis of DEGs between Purple Non-Heading Chinese Cabbage and Its Green Mutant

For comprehensive annotation, all DEGs were mapped to the GO. DEGs were significantly enriched in 27 GO terms: 7 terms for biological process, 5 terms for cellular component, and 15 terms for molecular function (Appendix A). The top-30 GO categories of the most significant enrichment are shown in Figure 4. *p* < 0.05 was considered as the threshold. In the biological process, “cellular glucan metabolic process” (GO:0006073), “glucan metabolic process” (GO:0044042), and “cellular polysaccharide metabolic process” (GO:0044264) were the most abundant terms. Under the cellular component category, “cell wall” (GO:0005618) and “external encapsulating structure” (GO:0030312) were significantly enriched. For the molecular function category, a large proportion of DEGs were related to “transferase activity, transferring hexosyl groups” (GO:0016758) and “hydrolase activity and hydrolyzing *O*-glycosyl compounds” (GO:0004553) (Figure 4 and Appendix A).

To further identify genes that participated in metabolic pathways, 287 DEGs were mapped to 89 KEGG pathways (Appendix A). DEGs were significantly enriched in the pathways of “phenylpropanoid biosynthesis” (brp00940), “flavonoid biosynthesis” (brp00941), “plant hormone signal transduction” (brp04075), “pentose and glucuronate interconversions” (brp00040), “phenylalanine metabolism” (brp00360), “DNA replication” (brp03030), “tyrosine metabolism” (brp00350), and “fatty acid degradation” (brp00071) (Figure 5 and Appendix A). Down-regulated DEGs were enriched in six pathways, including “DNA replication”, “pentose and glucuronate interconversions”, “phenylalanine metabolism”, “flavonoid biosynthesis”, “tyrosine metabolism”, and “phenylpropanoid biosynthesis”, of which “phenylpropanoid biosynthesis”, “flavonoid biosynthesis”, and “phenylalanine metabolism” were related to anthocyanin biosynthesis, and thirty-one, fourteen, and eleven DEGs were identified in these pathways, respectively, suggesting that the genes may be responsible for the different colors in the non-heading Chinese cabbage samples (Appendix A). Thirty-one DEGs associated with anthocyanin biosynthesis and regulation were identified. The results offer valuable information regarding the mechanism of anthocyanin accumulation in non-heading Chinese cabbage.

### 3.5. Anthocyanin Biosynthetic DEGs in Purple Non-Heading Chinese Cabbage and Its Green Mutant

The color of plant leaves depends on pigments in vacuoles, particularly anthocyanins. The RNA-seq results indicated that twenty-seven DEGs related to anthocyanin biosynthesis were identified in purple and green non-heading Chinese cabbage. Most of the secondary metabolite pathways were weakened by gene expression down-regulation in the green plants compared with purple plants, except for the DEGs *4CL* (*BraA05g025870.3C*) and *ACC* (*BraA06g004850.3C*), consistent with the high content of anthocyanins and the change in leaf color (Table 2 and Figure 6A). In non-heading Chinese cabbage, the phenylpropanoid pathway contained eight DEGs, including four *PAL*, three *4CL*, and one *ACC*. Four *PAL* genes (*BraA04g026260.3C*, *BraA09g044270.3C*, *BraA04g006280.3C*, and *BraA07g021160.3C*) were significantly down-regulated, two (*BraA07g031570.3C* and *BraA03g039690.3C*) of the three *4CL* genes were prominently down-regulated and one (*BraA05g025870.3C*) was significantly upregulated, and one *ACC* (*BraA06g004850.3C*) was significantly up-regulated in the green mutant compared with purple non-heading Chinese cabbage (Table 2 and Figure 6A). Simultaneous large-scale down-regulation of structural DEGs was not only identified in the phenylpropanoid pathway, but was also found in the flavonoid biosynthetic pathways. The early biosynthesis contained ten DEGs, including three *CHS* (*BraA10g024990.3C*, *BraA03g005990.3C*, and *BraA02g005190.3C*), one chalcone isomerase-like (*CHI-L*; *BraA10g028200.3C*), two *CHI* (*BraA09g046060.3C* and *BraA07g022020.3C*), two *F3H* (*BraA09g042420.3C* and *BraA03g045490.3C*), one *F3′H* (*BraA10g030360.3C*), and one *FLS* (*BraA06g027070.3C*) that dominated secondary metabolite synthesis modulation in the purple and green samples (Table 2 and Figure 6A). The expression of these ten DEGs was significantly higher in the purple sample compared with the green sample. High-fold down-regulation and low FPKM reduced the flux in the anthocyanidin biosynthetic pathways of green non-heading Chinese cabbage. The later biosynthesis included one *DFR* (*BraA09g019440.3C*), two *ANS* (*BraA01g013470.3C* and *BraA03g050560.3C*), one anthocyanidin 3-*O*-glucoside 5-*O*-glucosyltransferase (*UGT75C1*; *BraA08g009740.3C*), one auxin glycosyltransferase (*UGT84A2*; *BraA01g032130.3C*), one malonyl-CoA:anthocyanin 5-*O*-glucoside-6′-*O*-malonyltransferase (*5MAT*; *BraA09g003850.3C*), and one *A3GlcCouT* (*BraA08g035120.3C*). Simultaneous large-scale down-regulation of these later biosynthetic genes (LBGs) of the flavonoid pathways was identified in the green mutant, particularly *ANS* gene (*BraA03g050560.3C*) expression, which decreased more than that of the other genes. Two *TT19* genes (*BraA02g006880.3C* and *BraA10g022740.3C*) involved in anthocyanin transport were significantly down-regulated in green non-heading Chinese cabbage compared with purple non-heading Chinese cabbage (Table 2 and Figure 6A). The differentially accumulated metabolites and specifically accumulated metabolites in the anthocyanin biosynthetic pathway (ko00942) contained one delphinidin (specifically accumulated in the green mutant), seven cyanidin (one down-regulated and six specifically accumulated in purple WT), two petunidin (one up-regulated and one down-regulated), three peonidin (down-regulated and two specifically accumulated in purple WT), two pelargonidin (specifically accumulated in purple WT), and one malvidin (specifically accumulated in purple WT) (Figure 6B). The genes identified in anthocyanin biosynthesis are crucial. MYB and bHLH coordinately regulate the expression of structural genes, which promotes anthocyanin formation, leading to the formation of purple leaves.

### 3.6. Analysis of Transcriptome Factors

Anthocyanin biosynthesis is regulated by TFs in plants. In this study, 83 differentially expressed TFs were identified between purple non-heading Chinese cabbage and the green mutant, of which 41 differentially expressed TFs were up-regulated and 42 were down-regulated (Appendix A). The most abundant TFs were annotated as WRKY, bHLH, and MYB, and other TFs were also identified in the present study (Appendix A). WRKY was the maximum number of differentially expressed TFs between the green mutant and purple WT (nine up-regulated and three down-regulated). In eleven bHLH TFs, six were up-regulated and five were down-regulated; six MYBs were up-regulated and four MYBs were down-regulated, among which the down-regulated *TT8* (*BraA09g028560.3C*) and *EGL3* (*BraA09g013280.3C*) were homologs of *AtTT8* and *AtEGL3*, which were bHLH TF, which positively regulates anthocyanin biosynthesis (Table 2) [20]. Additionally, the *MYBL2* (*BraA07g035710.3C*) and *LBD39* (*BraA03g060820.3C*) genes exhibited higher expression levels in the purple cultivar than in the green mutant (Table 2). Two WD40s were down-regulated between the green and purple samples (Appendix A). These DEGs may be involved in the formation of different color leaves in non-heading Chinese cabbage.

### 3.7. Expression Patterns of the Genes Involved in Chlorophyll Biosynthesis

Critical genes associated with chlorophyll biosynthesis were identified. The FPKM values for *HEMB2* (*BraA10g007810.3C*), *PORA* (*BraA10g012360.3C*), and *CLH2* (*BraA09g019990.3C*) were <1 in the purple and green samples (Figure 7 and Appendix A). *HEMB1* (*BraA07g034610.3C*), *HEMG1* (*BraA10g001110.3C*), *CHLH* (*BraA03g005840.3C*), *CHLD* (*BraA06g005680.3C*), *CHLI* (*BraA01g009960.3C*), *CRD1* (*BraA07g022590.3C*), *PORB* (*BraA01g018190.3C* and *BraA03g053700.3C*), *PORC* (*BraA08g035300.3C* and *BraA10g002190.3C*), *CHLG* (*BraA01g022230.3C*), *CLD1* (*BraA07g019570.3C*), *HCAR* (*BraA10g003160.3C*), and NYC (*BraA04g008820.3C* and *BraA08g006610.3C*) had relatively high FPKM values (Figure 7 and Appendix A). The expression levels of *HEMD* (*BraA09g052610.3C*), *CHLH* (*BraA03g005840.3C*), *CRD1* (*BraA04g003690.3C* and *BraA07g022590.3C*), *CHLG* (*BraA01g022230.3C*), *CLD1* (*BraA07g019570.3C*), *CAO* (*BraA10g007770.3C*), *HCAR* (*BraA10g003160.3C*), *NOL* (*BraA03g001800.3C*), *CLH2* (*BraA09g019990.3C*), and *RCCR* (*BraA01g001750.3C*) in the green mutant were over one-fold higher than those in the purple WT, but there were no DEGs in Green vs. Purple.

### 3.8. Validation of the Transcriptomic Data by qRT-PCR

The transcription levels of thirty-one DEGs related to anthocyanin biosynthesis and regulation were examined in purple and green non-heading Chinese cabbage by qRT-PCR to confirm the RNA-Seq results. The results showed that four *PAL* (*BraA04g026260.3C*, *BraA09g044270.3C*, *BraA04g006280.3C*, and *BraA07g021160.3C*), two *4CL* (*BraA07g031570.3C* and *BraA03g039690.3C*), three *CHS* (*BraA10g024990.3C*, *BraA03g005990.3C*, and *BraA02g005190.3C*), two *CHI* (*BraA09g046060.3C* and *BraA07g022020.3C*,), one *CHI-L* (*BraA10g028200.3C*), two *F3H* (*BraA09g042420.3C* and *BraA03g045490.3C*), one *F3′H* (*BraA10g030360.3C*), one *FLS* (*BraA06g027070.3C*), one *DFR* (*BraA09g019440.3C*), two *ANS* (*BraA01g013470.3C* and *BraA03g050560.3C*), one *UGT75C1* (*BraA08g009740.3C*), one *UGT84A2* (*BraA01g032130.3C*), one *5MAT* (*BraA09g003850.3C*), one *A3GlcCouT* (*BraA08g035120.3C*), two *TT19* (*BraA02g006880.3C* and *BraA10g022740.3C*), one *TT8* (*BraA09g028560.3C*), one *EGL3* (*BraA09g013280.3C*), one *MYBL2* (*BraA07g035710.3C*), and one *LBD39* (*BraA03g060820.3C*) were up-regulated significantly in the purple non-heading Chinese cabbage compared with the expression in the green mutant (Figure 8). Significant up-regulation of *4CL* (*BraA05g025870.3C*) and *ACC* (*BraA06g004850.3C*) was observed in green non-heading Chinese cabbage (Figure 8). The results of qRT-PCR were comparable to the RNA-Seq analysis results and validated the reliability of transcriptome analysis (Figure 8 and Appendix A).

## 4. Discussion

Non-heading Chinese cabbage is an important vegetable crop because of its high nutritional value. Purple-leaved vegetables have attracted more attention because of their higher nutritional component contents, such as anthocyanins. Anthocyanin extracts from colored vegetables can be processed into functional components to produce food according to their potential health effects. In years past, anthocyanin biosynthesis attracted extensive attention in *Brassica* vegetables, such as heading Chinese cabbage (*Brassica rapa* L. ssp. *Pekinensis*), broccoli (*Brassica oleracea* L. Var. *Italica*), ornamental cabbage (*Brassica oleracea* var. *Acephala*), and mizuna (*Brassica rapa* L. Var. *Japonica*) [1,24,25,30]. It is important to study the reason for the formation of a purple leaf color in non-heading Chinese cabbage. In the present study, integrative metabolome and transcriptome analysis was applied to understand the formation of leaf color in non-heading Chinese cabbage.

The difference in leaf color is due to the different contents, distribution, biosynthesis, and types of anthocyanins, chlorophylls, and carotenoids [31]. The leaves of non-heading Chinese cabbage are purple and green, and the intracellular pigments had significant differences between the purple and green non-heading Chinese cabbage, indicating that they play an essential role in their distinct leaf colors. The leaf color of purple-leaved plants is closely associated with the types and proportions of pigments. The anthocyanin content of purple samples was higher than that of green samples, while the contents of chlorophyll and carotenoids were lower [32]; these results were also obtained in this study. There is an obvious negative correlation between the contents of anthocyanin and chlorophyll in colorful vegetables [10]. In the present study, the main pigment of the green mutant was chlorophyll, while anthocyanin was the key factor affecting purple non-heading Chinese cabbage. The expression level of most structural genes associated with chlorophyll biosynthesis in the green mutant was high compared with the purple WT, whereas high accumulation of anthocyanins corresponded with the expression level of genes related to anthocyanin biosynthesis in purple plants. These results were in line with those of a previous study indicating that chlorophyll is very important in photosynthesis and anthocyanins protect the photosynthetic system [21]. Anthocyanin is one of the most significant pigments that decide leaf color. Anthocyanin-related studies in *Brassica* crops have focused on the isolation and identification of metabolites [33]. In this regard, Chiu et al. found that the purple cauliflower (*Brassica oleracea* var *botrytis*) contained cyanidin 3-(coumaryl-caffeyl) glucoside-5-(malonyl)-glucoside using high-performance liquid chromatography (HPLC)-ESI-MS/MS analysis [34]. In addition, more than thirty cyanidin compounds have been found in red cabbage (*Brassica oleracea* L. Var. *capitata*) using HPLC–ESI–MS/MS [35]. Song et al. found that both the ratio of non-aromatic acylated cyanidin to aromatic acylated cyanidin and the ratio of the anthocyanin content to the chlorophyll content were responsible for the leaf color formation in different purple pakchoi lines [36]. In the present study, delphinidin, pelargonidin, petunidin, peonidin, and malvidin were also identified in purple non-heading Chinese cabbage besides cyanidin compared with purple mizuna [25]. Moreover, delphinidin-3-*O*-glucoside, delphinidin-3-*O*-galactoside, cyanidin-3-*O*-glucoside, cyanidin-3-*O*-galactoside, malvidin-3-*O*-glucoside, and malvidin-3-*O*-galactoside were identified in purple broccoli (*Brassica oleracea* L. var *Italica*) [37]. Similar results were found in this study—delphinidin-3-*O*-glucoside, delphinidin-3-*O*-galactoside, and cyanidin-3-*O*-galactoside were isolated from non-heading Chinese cabbage by LC-MS/MS. Furthermore, eleven metabolites (Table 1) were specially identified in the purple non-heading Chinese cabbage compared with purple broccoli. These findings showed that the significant differences in the anthocyanin secondary metabolites between the purple WT and green mutant may have been the reason for the formation of purple leaves in non-heading Chinese cabbage.

Anthocyanins are the final products of the flavonoid biosynthesis pathway. The biosynthesis of anthocyanin metabolites is controlled by structural genes. Yuan et al. reported that most of the structural genes in the anthocyanin biosynthesis pathway were more up-regulated during the vegetative growth of the red cabbage compared with the green cabbage and caused varying leaf colors [38]. Coordinated expression varieties of *F3H*, *F3′H*, *DFR*, *ANS*, and *UFGT* have been identified in differently colored *Arabidopsis*, mizuna, and other plants [15,26,39]. Similar results were observed in this study. There was a prominent increase in the transcriptional expression level of the structural genes in the phenylpropane and flavonoid biosynthesis pathways for the purple non-heading Chinese cabbage than in the green mutant (Table 2 and Figure 6), from the upstream *PAL* to the downstream *UGT*, which were determined by RNA-Seq and strongly supported our metabolomic results. DFR from various plants has a specific substrate bias for dihydrokaempferol, dihydromyricetin, and dihydroquercetin [39]. Furthermore, ANS, a key enzyme at the end of the anthocyanin synthesis pathway, catalyzes the conversion of colorless to colored anthocyanins [15]. In this study, *DFR* (*BraA09g019440.3C*) expression was significantly up-regulated in the purple sample compared with the green sample, which may have catalyzed the increased production of colorless anthocyanin metabolites in the purple variety (Figure 6). The most significant up-regulation of *ANS* (*BraA01g013470.3C* and *BraA03g050560.3C*) in structural genes may result in more colorless anthocyanin production from the catalysis of *DFR* into pelargonidin, cyanidin, and delphinidin, which is line with results of the higher amount of anthocyanin metabolites identified by the metabolome in the purple non-heading Chinese cabbage. Anthocyanins are extremely unstable and easily degradable. Glycosylation stabilizes anthocyanins and the anthocyanin transport signal to vacuoles. Thus, anthocyanins can play a role as pigments in vacuoles in the same ways as flavonoid 3-*O*-glucosyltransferase and anthocyanidin 3-*O*-glucosyltransferase-related genes [40,41]. Furthermore, Hiromoto et al. reported that anthocyanidin 3-*O*-glucosyltransferase (*UGT78K6*) catalyzed the conversion of delphinidin to delphinidin-3-*O*-glucoside in *Clitoria ternatea* [13]. In addition, Saito et al. claimed that *Fh3GT1* encoding *UF3GT* plays a crucial role in anthocyanin glycoside biosynthesis in *Freesia hybrida* [38], which was similar to the findings of this study. In purple non-heading Chinese cabbage, the accumulated pelargonidin-3-*O*-glucoside and cyanidin -3-*O*-glucoside were converted by up-regulating *UGT75C1* (*BraA08g009740.3C*) into pelargonidin-3,5-*O*-diglucoside and cyanidin-3,5-*O*-diglucoside (Figure 6), which are involved in the formation of purple leaves. The results of this study indicate that the high accumulation of metabolites in the anthocyanin biosynthesis pathway in purple non-heading Chinese cabbage might have been triggered by the high activity of the related structural genes.

The accumulation of chlorophyll plays an important role in the formation of plant leaf color. In the present study, a few structural genes that might be involved in chlorophyll biosynthesis and accumulation were identified, of which one-third of the genes had a higher expression level in the green mutant than in the purple WT, which is in line with the chlorophyll content (Figure 1 and Figure 7). Similar results were obtained in research on the gene regulation of chlorophyll biosynthesis in Ornamental Kale [10]. These genes might be involved in chlorophyll biosynthesis.

Numerous TFs are thought to regulate the transcriptional level of anthocyanin biosynthesis, including MYB, bHLH, WD40, LBD, and members of a few other TF families. TFs play a crucial role in regulating genes of the anthocyanin biosynthetic pathway or in the regulation of the single important genes in the color formation of vegetables, fruits, and flowers [2,3,35]. Previous studies indicated that WD-repeat/MYB/bHLH transcriptional complexes mainly regulated the late biosynthetic genes (LBGs) in the flavonoid biosynthesis pathway [20,42]. The enhanced content of flavonoid pigments is due to the up-regulation of the phenylpropanoid pathway, indicated by increases in the LBGs and early biosynthetic genes (EBGs) expressions in *PAP1*-overexpressing *Arabidopsis* leaves [43]. This observation suggests that transcriptional complexes in *Arabidopsis* possibly not only regulated LBGs, but also other genes in the flavonoid pathway. Similarly, the coordinated *MYB2* and *TT8* expression improved anthocyanin production through the up-regulation of late structural genes (*DFR*, *ANS*, *UFGT*, *GST*, and *LDOX*) and early structural genes (*CHS*, *F3H*, and *F3′H*) in purple heading Chinese cabbage and red cabbage, respectively [38,44]. Furthermore, the cooperation of MYB113 or MYB114 with TTG1- and bHLH (GL3, EGL3, and TT8) affected the expression of the *F3′H, DFR*, *LDOX, UGT75C1*, and *GST12* structural genes in *Arabidopsis* [20]. *BrEGL3.1*, *BrEGL3.2*, and *bHLH49* are candidate genes controlling anthocyanin accumulation in zicaitai (*Brassica rapa* L. ssp. *chinensis* var. *purpurea*) [44,45]; *BrEGL3.1* (*BraA09g015130.3C*) was not a DEG, but *EGL3.2* (*BraA09g013280.3C*) and *bHLH49* (*BraA07g033960.3C*) were more significantly upregulated in the purple non-heading Chinese cabbage than in the green mutant (Appendix A). *BrTT8* (*BraA09g028560.3C*) was significantly upregulated in the purple non-heading Chinese cabbage than in the green mutant (Table 2), which probably activates the regulation of anthocyanin biosynthesis in heading Chinese cabbage (*Brassica rapa* L. ssp. *pekinensis*) [46,47]. Furthermore, four MYB genes and two WD40 genes were down-regulated, which may positively regulate the expression of most structural DEGs related to anthocyanin biosynthesis with the MBW complex in purple non-heading Chinese cabbage. MYBL2 and LBD39 negatively regulate anthocyanin biosynthesis in *Arabidopsis* [21,23]. Furthermore, Song et al. found that the loss of *BoMYBL2-1* expression resulted in the production of the purple color in *B. oleracea* [47]. In contrast, our findings are contrary to these results. In the green mutant, the *MYBL2* (*BraA07g035710.3C*) and *LBD39* (*BraA03g060820.3C*) expressions were significantly down-regulated compared with the purple sample (Table 2). Therefore, MYBL2 and LBD39 may positively regulate anthocyanin accumulation in non-heading Chinese cabbage, and this inference needs to be further verified.

## 5. Conclusions

The formation of leaf color in non-heading Chinese cabbage was studied by integrative metabolome and transcriptome analysis. Changes in anthocyanin metabolites, particularly the specific accumulation of anthocyanins in the purple plant, underlined the color difference. Moreover, the DEGs and TFs related to anthocyanin biosynthesis were identified between purple non-heading Chinese cabbage and the green mutant. The crucial genes associated with chlorophyll biosynthesis were determined. These results offer valuable information regarding the formation of purple leaves in non-heading Chinese cabbage.

## Figures and Tables

**Figure 1 foods-11-01787-f001:**
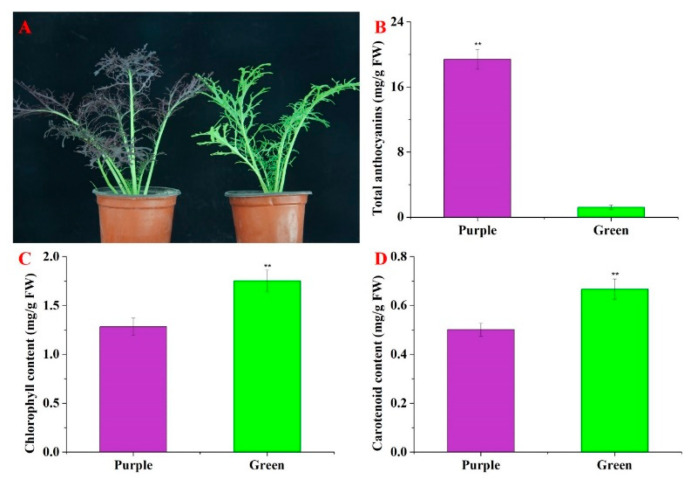
Phenotypes and anthocyanin content of the purple non-heading Chinese cabbage and its green mutant. (**A**) Phenotypes of purple non-heading Chinese cabbage and its green mutant. Bar = 1 cm. (**B**) The anthocyanin content in leaves of purple non-heading Chinese cabbage and its green mutant. (**C**) Chlorophyll content in the leaves of purple non-heading Chinese cabbage and its green mutant. (**D**) Carotenoid content in the leaves of purple non-heading Chinese cabbage and its green mutant. Error bars show standard errors (SE). ** indicates significant differences at *p* ≤ 0.01.

**Figure 2 foods-11-01787-f002:**
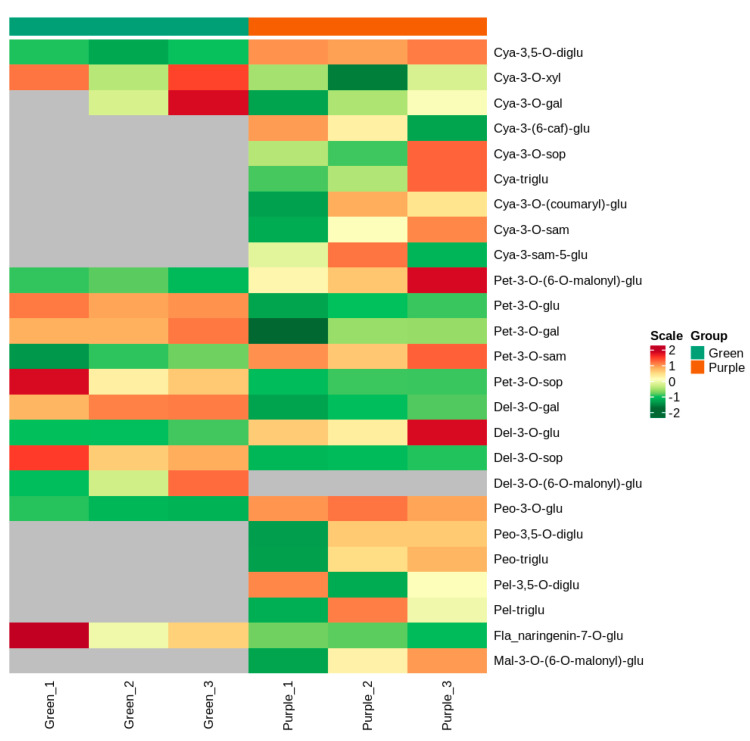
Cluster heatmap of metabolites between purple non-heading Chinese cabbage and its mutant. The x-axis shows sample information, the y-axis indicates the metabolite information, and the scale is the expression obtained after standardization.

**Figure 3 foods-11-01787-f003:**
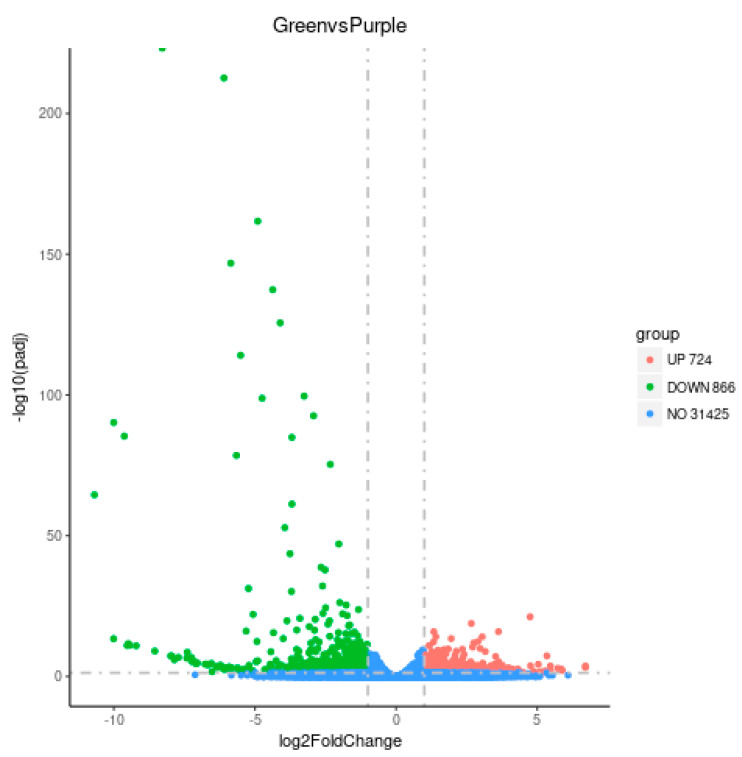
Identification of DEGs between purple non-heading Chinese cabbage and its green mutant.

**Figure 4 foods-11-01787-f004:**
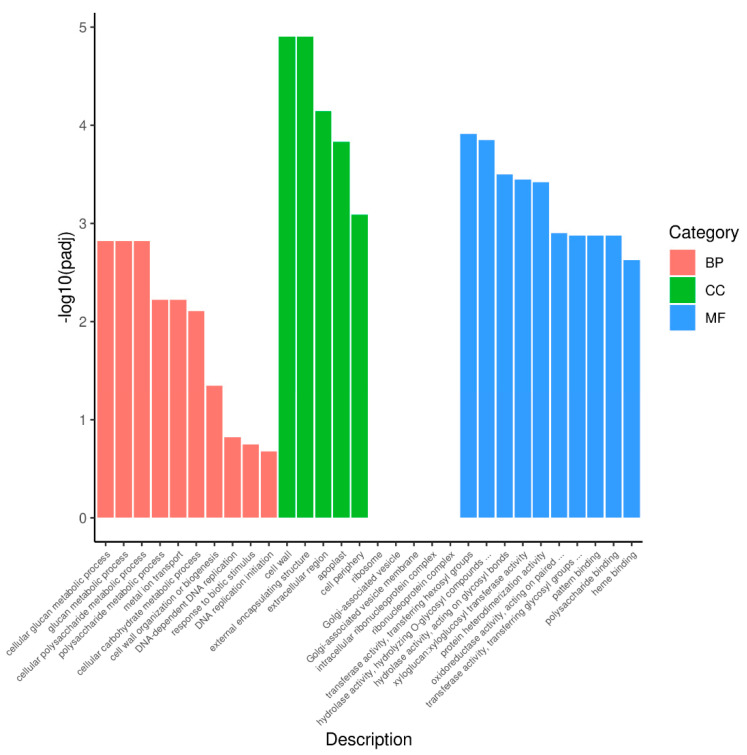
GO enrichment analysis of the DEGs between purple non-heading Chinese cabbage and its green mutant. BP, CC, and MF indicate the biological process, cellular component, and molecular function, respectively.

**Figure 5 foods-11-01787-f005:**
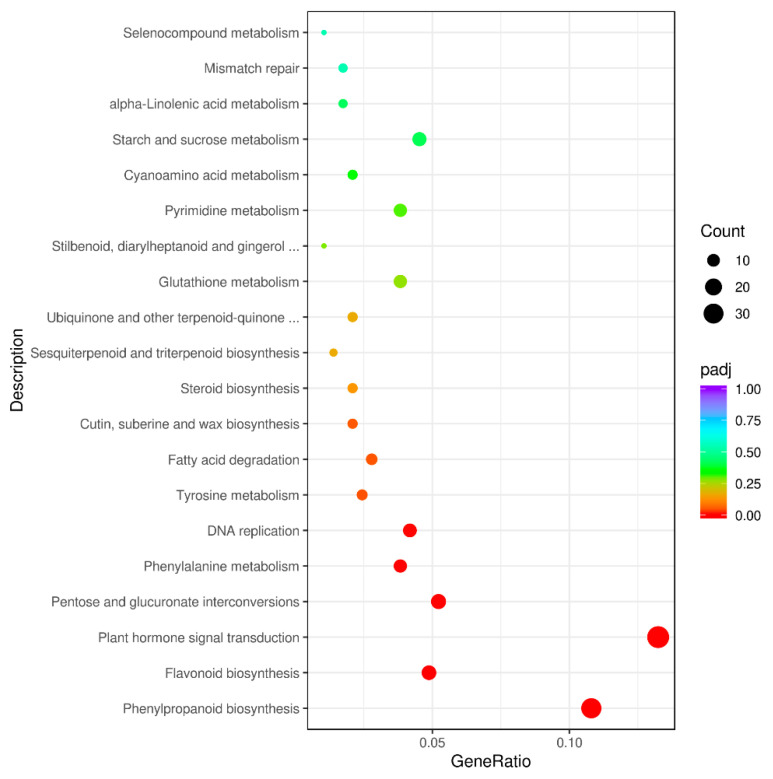
KEGG pathway analysis of the DEGs between purple non-heading Chinese cabbage and its green mutant.

**Figure 6 foods-11-01787-f006:**
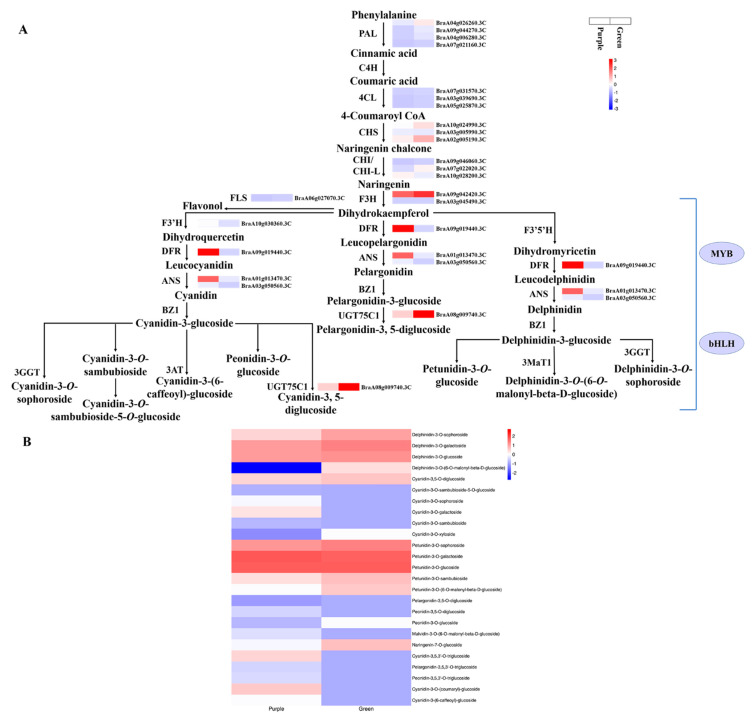
Transcript profiling of genes in the anthocyanin biosynthetic pathway in purple non-heading Chinese cabbage and its green mutant. (**A**) Reconstruction of anthocyanin biosynthetic pathway by differentially expressed structural genes and regulators. (**B**) Differentially accumulated metabolites in anthocyanin biosynthetic pathway. PAL, phenylalanine ammonia-lyase; C4H, cinnamate 4-hydroxylase; 4CL, 4-coumaroyl CoA ligase; CHS, chalcone synthase; CHI, chalcone isomerase; F3H, flavanone 3-hydroxylase; F3′H, flavanone 3′-hydroxylase; F3′5′H, flavanone 3′-5′-hydroxylase; DFR, dihydroflavonol 4-reductase; ANS, anthocyanidin synthase; UGT75C1, anthocyanidin 3-*O*-glucoside 5-*O*-glucosyltransferase; and BZ1, anthocyanin Bronze-1.

**Figure 7 foods-11-01787-f007:**
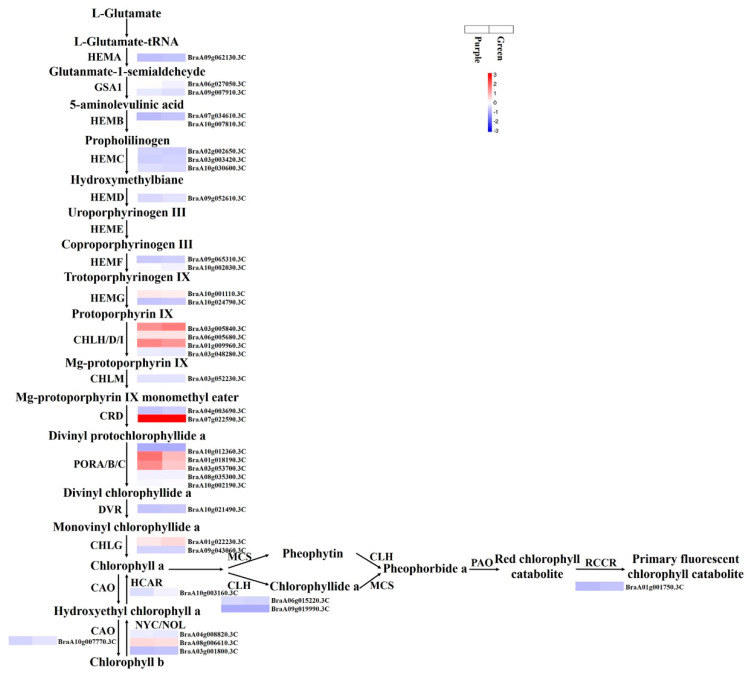
Chlorophyll biosynthetic pathway in non-heading Chinese cabbage.

**Figure 8 foods-11-01787-f008:**
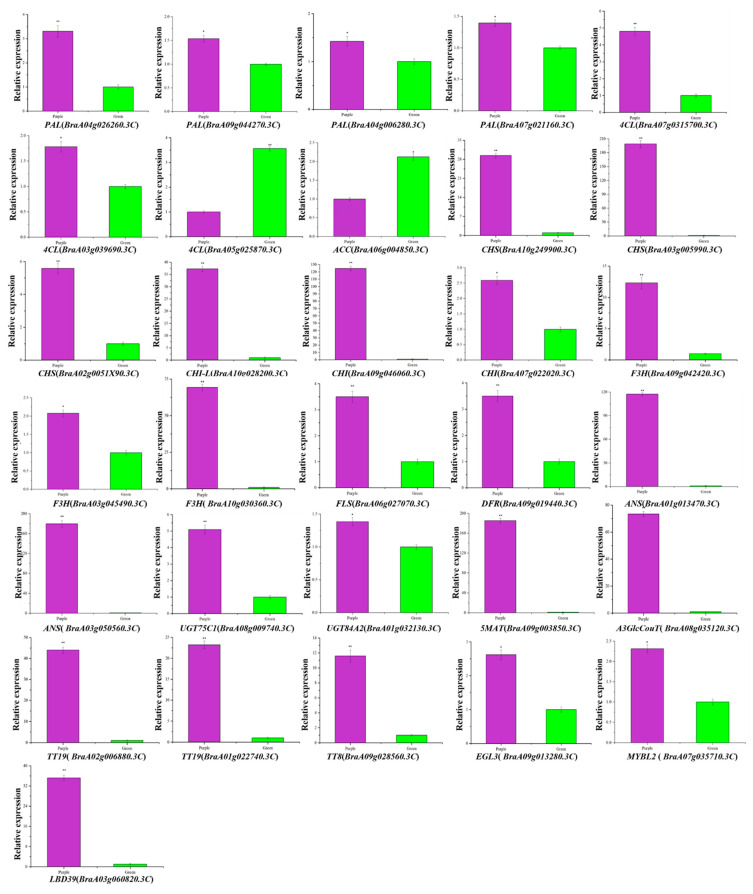
Expression analysis of DEGs associated with anthocyanin biosynthesis between purple non-heading Chinese cabbage and its green mutant. Expression analysis of *PAL* (*BraA04g026260.3C*, *BraA09g044270.3C*, *BraA04g006280.3C*, and *BraA07g021160.3C*), three *4CL* (*BraA07g031570.3C*, *BraA03g039690.3C*, and *BraA05g025870.3C*), one *ACC* (*BraA06g004850.3C*), three *CHS* (*BraA10g024990.3C*, *BraA03g005990.3C*, and *BraA02g005190.3C*), two *CHI* (*BraA09g046060.3C* and *BraA07g022020.3C*,), one *CHI-L* (*BraA10g028200.3C*), two *F3H* (*BraA09g042420.3C* and *BraA03g045490.3C*), one *F3′H* (*BraA10g030360.3C*), one *FLS* (*BraA06g027070.3C*), one *DFR* (*BraA09g019440.3C*), two *ANS* (*BraA01g013470.3C* and *BraA03g050560.3C*), one *UGT75C1* (*BraA08g009740.3C*), one *UGT84A2* (*BraA01g032130.3C*), one *5MAT* (*BraA09g003850.3C*), one *A3GlcCouT* (*BraA08g035120.3C*), two *TT19* (*BraA02g006880.3C* and *BraA10g022740.3C*), one *TT8* (*BraA09g028560.3C*), one *EGL3* (*BraA09g013280.3C*), one *MYBL2* (*BraA07g035710.3C*), and one *LBD39* (*BraA03g060820.3C*) in purple non-heading Chinese cabbage and its green mutant. Error bars represent the standard error (SE). ** indicates significant differences at *p* ≤ 0.01 and * indicates significant differences at *p* ≤ 0.05.

**Table 1 foods-11-01787-t001:** The contents of anthocyanin metabolites detected in purple non-heading Chinese cabbage and its green mutant.

Compounds	Content in Purple (µg/g)		Content in Green (µg/g)		Fold Change
	Mean	SE	Mean	SE	
Delphinidin-3-*O*-sophoroside	3.3800	0.0617	6.4200	0.3990	1.9000
Delphinidin-3-*O*-galactoside	26.4000	1.3500	48.6000	1.1900	1.8400
Delphinidin-3-*O*-glucoside	27.8000	2.5900	17.2000	0.1860	0.6190
Delphinidin-3-*O*-(6-O-malonyl-beta-D-glucoside)	N/A	N/A	0.129	0.0044	N/A
Cyanidin-3,5-*O*-diglucoside	3.3300	0.0814	0.5810	0.1200	0.1750
Cyanidin-3-*O*-sambubioside-5-*O*-glucoside	0.0316	0.0002	N/A	N/A	N/A
Cyanidin-3-*O*-sophoroside	0.4870	0.0155	N/A	N/A	N/A
Cyanidin-3-*O*-galactoside	1.9100	0.1540	2.4600	0.4050	1.2800
Cyanidin-3-*O*-sambubioside	0.0385	0.0022	N/A	N/A	N/A
Cyanidin-3-*O*-xyloside	0.0068	0.0006	0.0091	0.0009	1.3500
Cyanidin-3-*O*-(coumaryl)-glucoside	4.8400	0.3150	N/A	N/A	N/A
Cyanidin-3-(6-caffeoyl)-glucoside	0.5440	0.0175	N/A	N/A	N/A
Cyanidin-3,5,3’-*O*-triglucoside	3.0200	0.0764	N/A	N/A	N/A
Petunidin-3-*O*-sophoroside	31.9000	0.6510	59.70000	6.9300	1.8700
Petunidin-3-*O*-galactoside	377.0000	15.8000	454.0000	4.3300	1.2100
Petunidin-3-*O*-glucoside	284.0000	15.9000	577.0000	10.1000	2.0300
Petunidin-3-*O*-sambubioside	2.0100	0.1160	0.7390	0.1190	0.3670
Petunidin-3-*O*-(6-*O*-malonyl-beta-D-glucoside)	0.6030	0.0523	0.4020	0.0076	0.6670
Peonidin-3,5,3’-*O*-triglucoside	0.1320	0.0044	N/A	N/A	N/A
Peonidin-3,5-*O*-diglucoside	0.1170	0.0030	N/A	N/A	N/A
Peonidin-3-*O*-glucoside	0.0350	0.0009	0.0109	0.0007	0.3110
Malvidin-3-*O*-(6-*O*-malonyl-beta-D-glucoside)	0.1670	0.0056	N/A	N/A	N/A
Naringenin-7-*O*-glucoside	0.4640	0.0214	0.8780	0.1420	1.8900
Pelargonidin-3,5-*O*-diglucoside	0.0128	0.0006	N/A	N/A	N/A
Pelargonidin-3,5,3’-*O*-triglucoside	0.1110	0.0006	N/A	N/A	N/A

Note: N/A means that the substance was not detected in this project.

**Table 2 foods-11-01787-t002:** DEGs related to anthocyanin biosynthesis and regulation between purple non-heading Chinese Cabbage and its green mutant.

Gene ID	Gene Name	Average FPKM of Purple	Average FPKM of Green	Log2 Fold Change	*p* Value	Padj
*BraA04g026260.3C*	*PAL*	123.39	33.22	−1.89	6.59 × 10^−26^	6.07 × 10^−23^
*BraA09g044270.3C*	*PAL*	31.13	11.23	−1.47	3.30 × 10^−19^	1.92 × 10^−16^
*BraA04g006280.3C*	*PAL*	25.71	9.26	−1.48	4.66 × 10^−15^	1.80 × 10^−12^
*BraA07g021160.3C*	*PAL*	1.42	0.58	−1.28	1.19 × 10^−3^	1.44 × 10^−2^
*BraA07g031570.3C*	*4CL*	18.39	3.47	−2.41	3.67 × 10^−22^	2.62 × 10^−19^
*BraA03g039690.3C*	*4CL*	4.28	1.50	−1.52	1.66 × 10^−4^	3.14 × 10^−3^
*BraA05g025870.3C*	*4CL*	0.01	0.25	4.41	5.38 × 10^−3^	4.43 × 10^−2^
*BraA06g004850.3C*	*ACC*	0.02	0.21	3.13	1.33 × 10^−5^	4.40 × 10^−4^
*BraA10g024990.3C*	*CHS*	208.44	37.86	−2.46	3.81 × 10^−7^	2.53 × 10^−5^
*BraA03g005990.3C*	*CHS*	154.68	11.44	−3.76	1.35 × 10^−5^	4.44 × 10^−4^
*BraA02g005190.3C*	*CHS*	297.88	58.44	−2.35	4.82 × 10^−8^	4.65 × 10^−6^
*BraA09g046060.3C*	*CHI*	3.56	0	−7.94	4.01 × 10^−10^	6.53 × 10^−8^
*BraA07g022020.3C*	*CHI*	59.44	28.25	−1.08	2.50 × 10^−4^	4.28 × 10^−3^
*BraA10g028200.3C*	*CHI-L*	281.65	13.87	−4.35	5.12 × 10^−19^	2.86 × 10^−16^
*BraA09g042420.3C*		854.91	112.69	−2.92	9.14 × 10^−97^	2.61 × 10^−93^
*BraA03g045490.3C*	*F3H*	21.68	1.43	−3.94	8.93 × 10^−57^	1.41 × 10^−53^
*BraA10g030360.3C*	*F3’H*	215.68	7.08	−4.93	9.41 × 10^−16^	4.04 × 10^−13^
*BraA06g027070.3C*	*FLS*	1.55	0.22	−2.80	1.57 × 10^−4^	3.01 × 10^−3^
*BraA09g019440.3C*	*DFR*	1194.67	3.87	−8.28	0	0
*BraA01g013470.3C*	*ANS*	869.00	12.68	−6.10	1.75 × 10^−217^	2.49 × 10^−213^
*BraA03g050560.3C*	*ANS*	147.04	0.19	−9.62	1.90 × 10^−89^	4.53 × 10^−86^
*BraA08g009740.3C*	*UGT75C1*	428.73	137.21	−1.64	7.75 × 10^−22^	5.40 × 10^−19^
*BraA01g032130.3C*	*UGT84A2*	21.56	6.75	−1.68	1.56 × 10^−21^	1.06 × 10^−18^
*BraA09g003850.3C*	*5MAT*	145.89	0.09	−10.69	1.66 × 10^−68^	2.95 × 10^−65^
*BraA08g035120.3C*	*A3GlcCouT*	323.27	5.60	−5.86	2.19 × 10^−151^	1.56 × 10^−147^
*BraA02g006880.3C*	*TT19*	567.41	99.37	−2.51	1.09 × 10^−41^	1.42 × 10^−38^
*BraA10g022740.3C*	*TT19*	457.80	22.22	−4.37	6.96 × 10^−142^	3.97 × 10^−138^
*BraA09g028560.3C*	*TT8*	56.12	1.88	−4.90	1.98 × 10^−166^	1.88 × 10^−162^
*BraA09g013280.3C*	*EGL3*	1.33	0.42	−1.67	1.01 × 10^−3^	1.28 × 10^−2^
*BraA07g035710.3C*	*MYBL2*	128.88	30.63	−2.08	2.04 × 10^−16^	9.38 × 10^−14^
*BraA03g060820.3C*	*LBD39*	4.69	0.66	−2.83	2.51 × 10^−6^	1.17 × 10^−4^

## Data Availability

The raw transcriptome sequencing data are available in the SRA database of the National Center for Biotechnology Information under the accession number of PRJNA789727 (www.ncbi.nlm.nih.gov/bioproject/PRJNA789727 (12 March 2022).

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
