# Peer review of "Transcriptome and Metabolome Profiling to Explore the Causes of Purple Leaves Formation in Non-Heading Chinese Cabbage (Brassica rapa L. ssp. chinensis Makino var. mutliceps Hort.)"

_foods, 2022, doi:10.3390/foods11121787_

Round 1

Reviewer 1 Report

The current  manuscript describes the transcriptomics and metabolomics defined mechanisms of the color generation for the non-heading Chinese cabbage. The work is substantial well designed and executed, and the presentation of the results is clear. Nevertheless, the metabolomics part is rather obscure in term of the procedures followed for the experimental part and the treatment of the data.

The comments can be found below

line 73  please rewrite this sentence as it seems difficult to comprehend correctly 

line 108 whirlpool:  is it vortexed? 

line 110 formulas  please refer how this formulas have been derived 

line 117 buffer  this not a buffer with the strict sense, the buffers are made of the conjugate acids or bases.

line 136  please refer the city and country of the vendor 

line 143  the authors need to clarify the procedure thorougly. Did they use a suspect list or selected MRM's? The use of databases is tailored for untargeted analyses, but the authors are using a low
res instrument. The workflow is confusing please provide many more details

table 4  Please identify what the units are referring to. Also please check out the significant digits and the powers of 10 i.e. 3.38E+00 is just 3.38 

figure 4  please annotate the legend accordingly 

Author Response

Reviewer 1:

Comments and Suggestions for Authors

The current manuscript describes the transcriptomics and metabolomics defined mechanisms of the color generation for the non-heading Chinese cabbage. The work is substantial well designed and executed, and the presentation of the results is clear. Nevertheless, the metabolomics part is rather obscure in term of the procedures followed for the experimental part and the treatment of the data.

The main corrections and responses are as follows:

1. line 73  please rewrite this sentence as it seems difficult to comprehend correctly 

Response: The sentence has been rewritten in line 69-71.

2. line 108 whirlpool:  is it vortexed? 

Response: We have changed ‘whirlpool’ to ‘vortexed’ in line 103.

3. line 110 formulas  please refer how this formulas have been derived 

Response: The formulas were cited from the Reference Li et al. (2021) in lines 101-102.

4. line 117 buffer  this not a buffer with the strict sense, the buffers are made of the conjugate acids or bases.

Response: We have changed ‘buffer’ to ‘solution’ in line 111.

5. line 136  please refer the city and country of the vendor 

Response: The city and country of the vendor has been added in lines 130-134.

6. line 143  the authors need to clarify the procedure thorougly. Did they use a suspect list or selected MRM's? The use of databases is tailored for untargeted analyses, but the authors are using a low res instrument. The workflow is confusing please provide many more details

Response: Anthocyanin-targeted metabolome analysis was performed at Wuhan MetWare Biotechnology Co., Ltd. The procedure has been described in lines 140-156. Quantitative analysis is completed by MRM of triple quadrupole linear ion trap mass spectrometer. The previously described database used for untargeted analyses is inaccurate, the relevant contents have been deleted and the database used in this study was MWDB (Metware Database) based on the standard for anthocyanin-targeted analyses (line 149). The workflow with detailed information has been added in lines 133-137.

7. table 4  Please identify what the units are referring to. Also please check out the significant digits and the powers of 10 i.e. 3.38E+00 is just 3.38 

Response: The unit of anthocyanin content is µg/g and this information has been added in Table 1. The significant digits have been checked in Table 1 and the powers has been changed.

8. figure 4  please annotate the legend accordingly 

Response: The legend of figure 4 have been annotated in lines 271-272.

Reviewer 2 Report

I have serious concern about the novelty and the interest of this paper for a journal with high impact factor like Foods.

The authors claimed that line 80: "there are few studies on genes associated with metabolic traits of anthocyanin biosynthesis in purple non-heading Chinese cabbage", which is not true. On the contrary there are very important studies dealing with this topic: https://doi.org/10.1021/acs.jafc.5b04674; https://doi.org/10.3390/horticulturae8050351; https://doi.org/10.1016/j.hpj.2016.11.007, to cite only few.

The transcriptomics results are in my opinion obvious and already described for many plant species. These results have to be explained in the present context, and to validate them for possible applications for non-heading Chinese cabbage.

The presentation of the phytochemical analysis is also not satisfactory for publication. Revise the Table 1 by using the same unit and format (please also indicate the unit clearly in the Table legend or footer).

Author Response

Reviewer 2:

Comments and Suggestions for Authors

1. I have serious concern about the novelty and the interest of this paper for a journal with high impact factor like Foods.

Response: In this study, the wide type was a purple DH line obtained from a purple non-heading Chinese cabbage strain by isolated microspore culture. A green leaf color mutant was spontaneously mutated from the self-pollination of this purple DH line. So their genetic backgrounds were the same except for their purple and green leaf color. Therefore, the mutant and its wide type are ideal materials for the transcriptomic analysis. The results provide some valuable information for elucidating the causes of purple leaf formation in non-heading Chinese cabbage.

2. The authors claimed that line 80: "there are few studies on genes associated with metabolic traits of anthocyanin biosynthesis in purple non-heading Chinese cabbage", which is not true. On the contrary there are very important studies dealing with this topic:https://doi.org/10.1021/acs.jafc.5b04674; https://doi.org/10.3390/horticulturae8050351; https://doi.org/10.1016/j.hpj.2016.11.007, to cite only few.

Response: We have rewritten this sentence and added the related information of the research about anthocyanin synthesis genes in purple Chinese cabbage in Introduction in lines 74-79.

3. The transcriptomics results are in my opinion obvious and already described for many plant species. These results have to be explained in the present context, and to validate them for possible applications for non-heading Chinese cabbage.

Response: Although transcriptomics results of anthocyanin biosynthetic are already described for many plant species, this study still provided some valuable information for purple leaf formation in non-heading Chinese cabbage. 27 anthocyanin biosynthetic genes and 83 transcription factors were significantly differentially expressed between green leaf mutant and its WT. Moreover, transcriptome and metabolome analyses showed that UGT75C1 catalyzing the formation of pelargonidin-3,5-O-diglucoside and cyanidin-3,5-O-diglucoside may play a critical role for purple leaf formation in non-heading Chinese cabbage. The comparison of our transcriptomics results were compared to other studies and it was added in the discussion in lines 505-512.These results have provided crucial information for elucidating the formation of purple leaves in non-heading Chinese cabbage.

4. The presentation of the phytochemical analysis is also not satisfactory for publication. Revise the Table 1 by using the same unit and format (please also indicate the unit clearly in the Table legend or footer).

Response: The phytochemical analysis has been redescribed in lines 216-229 and Table 1 has been revised using the same unit and format.

Round 2

Reviewer 2 Report

The authors have replied to all my concerns and questions.